# Is the Frequency of Candidemia Increasing in COVID-19 Patients Receiving Corticosteroids?

**DOI:** 10.3390/jof6040286

**Published:** 2020-11-13

**Authors:** Cezar V. W. Riche, Renato Cassol, Alessandro C. Pasqualotto

**Affiliations:** 1Hospital Ernesto Dornelles, Porto Alegre 90160-092, Brazil; cezar.riche@gmail.com; 2Hospital Nossa Senhora da Conceição, Porto Alegre 91350-200, Brazil; renato.cassol@gmail.com; 3School of Medicine, Universidade Federal de Ciências da Saúde de Porto Alegre, Porto Alegre 90050-170, Brazil; 4Santa Casa de Misericordia de Porto Alegre, Porto Alegre 90020-090, Brazil

**Keywords:** COVID-19, candidemia, steroids

## Abstract

Corticosteroids have potent anti-inflammatory and immunosuppressive effects. Recently, these medications have gained importance in the treatment of severe COVID-19. Here we present data demonstrating a marked (10-fold) increase in frequency of candidemia in hospitalized patients with COVID-19 receiving corticosteroids in Brazil. Overall mortality was 72.7%, despite antifungal therapy. Physicians should be aware of the potential risk for candidemia among severely ill COVID-19 patients receiving high-doses of corticosteroids.

## 1. Introduction

Recent trials have demonstrated the benefit of corticosteroids in the treatment of severe COVID-19 [1,2,3,4], which is likely to result in a marked increase in steroids prescriptions in clinical practice. Steroids are known to increase the risks for a variety of infections, but data on serious fungal diseases in critically ill patients receiving corticosteroids for COVID-19 remains scarce. In the Randomized, Embedded, Multifactorial, Adaptive Platform trial for Community-Acquired Pneumonia (REMAP-CAP), despite the benefits observed with hydrocortisone treatment, the authors demonstrated a higher number of serious adverse events in both hydrocortisone groups [2], even though only one patient had documented fungemia in the trial.

The relationship between corticosteroid treatment and fungal infections is well established. Corticosteroids may promote fungal growth in vitro [5]. Moreover, use of corticosteroids have been associated with increased risk for most serious fungal diseases including candidemia, invasive aspergillosis, fusariosis, and mucormycosis [6]. He we present provocative results of a series of candidemia cases following COVID-19, in which the use of steroids was suggested as the main risk factor for fungal infection.

## 2. Materials and Methods

This was a case series of candidemia following COVID-19 in two tertiary care hospitals located in Porto Alegre, Southern Brazil. COVID-19 was diagnosed based on a positive real time PCR test in a patient with appropriate signs and symptoms, according to the criteria established by the Brazilian National as well as the Regional Sanitary Surveillance agencies [7,8]. Candidemia was defined as the presence of one or more *Candida* species in blood cultures, in patients with temporally clinical manifestations. Patients were included only once in the study. Yeasts were identified at the species level either by Phoenix Automated Microbiology System (Becton Dickinson, Franklin Lakes, NJ, USA) or Vitek-2 System (bioMérieux, Marcy-l’Étoile, France).

Data were retrospectively collected in the year 2020 between March 16 (date of the first COVID-19 admission in the participant hospitals) and August 31. Data were obtained from infection control records, as well as patients’ electronic charts. The Chi-square test with Yates correction was used to compare proportions—WINPEPI v. 11.65 (Abramson, JH) [9], and *p* values of < 5% were considered statistically significant.

## 3. Results

Since the beginning of the COVID-19 pandemic, an increase in the frequency of candidemia was observed in two of the hospitals in which authors (CVWR, and RC) were based. During the period of study, incidence of candidemia was 1.43 (hospital 1) and 1.15 (hospital 2) in patients who did not have COVID-19, both per 1000 patients-day. These frequencies were unaltered in comparison to previous year. In contrast, the incidence of candidemia in patients with COVID-19 were 11.83 (hospital 1) and 10.23 (hospital 2) per 1000 patients-day, during the same period (*p* = 0.001 in comparison to the prevalence of candidemia in non-COVID-19 patients). In patients with COVID-19, all cases of candidemia occurred after the use of high-doses of corticosteroids for severe disease. Table 1 summarizes the main clinical findings for these 11 patients. These were mostly male (*n* = 7; 63.6%), with a median age of 59 years-old (interquartile range, IQR, 49–70 years-old), and a median Charlson score of 4 (IQR, 1–4). Four patients had diabetes, and one was HIV-positive. Ten patients (90.9%) had central venous catheters in place. These were all non-surgical patients with limited antibiotic exposure, and additional risk factors for candidemia were virtually absent. Candidemia occurred in intensive care for 8/11 patients (72.7%), and after a median length of admission to the intensive care unit of 8.7 days. Overall mortality of candidemia following steroid use in COVID-19 was 72.7% (8/11), while 3/11 patients (22.3%) were discharged from the hospital. Overall mortality in COVID-19 patients who did not develop candidemia was 17.7% (hospital 1) and 22.0% (hospital 2).

## 4. Discussion

In this small case series, we observed a 10-fold increase in the frequency of candidemia in two medical centers in Brazil, in COVID-19 patients taking high doses of steroids. The in-hospital mortality among COVID-19 patients in the study was similar to the published findings of others [10], and even lower than other studies conducted in the same region in Brazil [11]. He observed an impressively high mortality in these patients. It should be noted, however, that our study is limited—mostly by its small sample size, in addition to its retrospective design and its short duration of observation.

Current data suggest low frequency of both bacterial and fungal infections in COVID-19 cases presenting early at the hospital [12]. However, the use of steroids may dramatically change such scenarios. Fungal infections are known to occur after COVID-19, with most studies so far focusing on invasive aspergillosis [13]. In our perspective, the real frequencies and organisms associated with COVID-19 superinfections must be further evaluated and might be increased by immunosuppressive treatments.

In conclusion, steroids are a lifesaving treatment option in COVID-19 pneumonia, but the right dose and timing of such interventions remain to be determined. High corticosteroid doses might be associated with severe superinfections. Physicians should be aware of the potential increase in the frequency of candidemia in patients with COVID-19 treated with corticosteroids. 

## Figures and Tables

**Table 1 jof-06-00286-t001:** Characteristics of patients who had candidemia following corticosteroid use for severe COVID-19.

Pt	Age/Sex	Underlying Conditions	Date of Admission	COVID-19 Management	ICU Days until Isolation/*Candida* Species	Risk Factors for Candidemia	Treatment and Outcome
1	44/M	DM, dyslipidemia, psoriasis, schizophrenia	13 April 2020	Chloroquine 4 d AZI, IVE MPD 1 g qd 2 d, then 80 mg bid 9 d and reduction	No ICU admission *C. albicans*	DM CVC 20 days PTZ 5 days MEM 8 days	AFG Discharged
2	71/F	HT, coronary artery disease	14 May 2020	AZI MPD 62.5 mg bid 2 d	1 day *C. glabrata*	CVC 7 days Ceftriaxone 2 d. PTZ 7 days	Death prior to treatment
3	69/F	DM, HT, obesity, cardiac failure, COPD	16 May 2020	HCQ 1 day ^a^/AZI MPD 62.5 mg tid 7 d	9 days *C. glabrata*	DM, CVC 9 days Oxacillin 4 days PTZ 7 days	AFG Death
4	37/F	Asthma	1 July 2020	AZI, IVE MPD 62.5 mg tid 2 d., then DXM 6 mg qd 3 d., MPD 62.5 mg quad 8 d	8 days *C. albicans*	CVC 8 days AMC 5 days PTZ 7 days	AMB-d FLC Death
5	87/F	HT, atrial fibrillation	10 July 2020	AZI, IVE MPD 62.5 mg tid 2 d., bid 2 d., and qd 3 d	6 days *C. tropicalis*	CVC 7 days PTZ 6 days	AFG Death
6	66/M	HT	26 July 2020	AZI, IVE MPD 1 g qd 2 d., then 500 mg qd 1 day, DXM 6 mg qd 9 d	14 days *C. albicans*	CVC 14 days Cefuroxime 5 d. PTZ 8 days MEM 3 days	AFG Discharged
7	91/M	Asthma	29 July 2020	MPD 250 mg qd 1 d., then DXM 6 mg qd 7 d	7 days *C. albicans*	CVC 7 days Amoxicillin 5 d. PTZ 7 days	AFG Death
8	54/M	DM, HT, obesity, Raynaud synd., depression	8 August 2020	AZI DXM 6 mg qd 5 d	Day of ICU admission *C. albicans*	DM AMC 6 days PTZ 2 days	FLC Discharged
9	41/M	HIV, HCV, myocardiopathy	15 August 2020	Prednisone 40 mg qd 4 d	Day of ICU admission *C. albicans*	HIV CVC 10 days	FLC Death
10	57/M	DM, HT	23 July 2020 ^b^	AZI DXM 6 mg qd 10 d	22 days *C. albicans*	DM CVC 22 days AMC 5 days PTZ 8 days	Voriconazole AFG Death
11	59/M	HT, chronic renal disease, mental impairment	23 August 2020	DXM 6 mg qd 3 d., then Hydrocortisone 50 mg qid 4 d	3 days *C. albicans*	CVC 4 days AMC 4 days	FLC Death

Abbreviations: F, female; M, male; Pt, patient; COPD, chronic obstructive pulmonary disease; DM, Diabetes mellitus; HCV, hepatitis C virus; HT, hypertension; AZI, Azithromycin; DMX, dexamethasone; days, d.; HQC, hydroxychloroquine; IVE, Ivermectin; ICU, intensive care unit; MPD, methylprednisolone; AMC, Amoxicillin/clavulanate; AMB-d, Amphotericin B deoxycholate; AFG, Anidulafungin; CVC central venous catheter; FLC, Fluconazole; MEM, Meropenem; PTZ, Piperacillin/tazobactam. ^a^ Hydroxychloroquine was stopped because of cardiac arrythmia. ^b^ COVID-19 symptoms began after hospital admission.

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
