# Peer review of "Is the Frequency of Candidemia Increasing in COVID-19 Patients Receiving Corticosteroids?"

_jof, 2020, doi:10.3390/jof6040286_

Round 1
Reviewer 1 Report
Line 10: consider writing "in the treatment of hospitalized patients" instead of "for hospitalized patients".
Line 11: should be COVID-19 with uppercase letters.
Line 11: should be COVID-19 disease not COVID-19.
Line 12-13: consider writing "However an important issue poorly examined is the” instead of “One aspect however that seems to have been poorly discussed".
Line 13: consider writing "under these treatment" instead of "taking these medications".
Line 13-14: should be "The relation between corticosteroid treatment and fungal infections is well established." instead of "the association between corticosteroids and fungal diseases is well known."
Line 15: grammatical error.
Line 16: This is the first time you mention REMAP-CAP trial. Define this: "Randomized, Embedded, Multifactorial, Adaptive Platform trial for Community-Acquired Pneumonia (REMAP-CAP)".
Line16-18: Sentence is too long. Consider breaking it up into two sentences.
Line 20: should be COVID-19 with uppercase letters.
Line 23: should be COVID-19 with uppercase letters.
Line 25-26: should be "During our study period, XX were male (55%) and XX were female (XX with an average age of 57 years (interquartile range, IQR, 44-69 years-old), and with median Charlson score of 4 (IQR, 1-4). Of the included patients, XX patients were diagnosed with diabetes, XX was HIV-positive. XX patients (88.9%) had central venous catheters in place." instead of "These patients were predominantly male (55%), median age was 57 years-old (interquartile range, IQR, 44-69 years-old), and median Charlson score was 4 (IQR, 1-4). Four patients had diabetes and one was infected with the HIV. Central venous catheters were in place for 88.9% (n=8) of patients."
Line 29: should be COVID-19 with capital letters.
Line 28-30: This is a long sentence. Consider breaking it up in two sentences. If not, check the grammar. I suspect there is an inconsistency between plural vs. singular and verb conjugations.
Line 38: should be COVID-19 with capital letters.
Table 1: Treatment and outcome column should be split in two different ones.
Author Response
Reviewer 1
Comments and Suggestions for Authors
Line 10: consider writing "in the treatment of hospitalized patients" instead of "for hospitalized patients".
We agree and modified the text, accordingly.
Lines 11, 20, 23, 29 and 38: should be COVID-19 with uppercase letters.
All were modified in the text.
Line 11: should be COVID-19 disease not COVID-19.
We have preffered to use COVID-19 as a synonym for “coronavirus disease”, as stated by the World Health Organization. Therefore, using “COVID-19 disease” would sound like reduncancy.
Line 12-13: consider writing "However an important issue poorly examined is the” instead of “One aspect however that seems to have been poorly discussed".
This sentence was modified as other aspects in the introduction.
Line 13: consider writing "under these treatment" instead of "taking these medications".
This sentence was also modified as other aspects in the introduction.
Line 13-14: should be "The relation between corticosteroid treatment and fungal infections is well established." instead of "the association between corticosteroids and fungal diseases is well known."
We agree and modified the text, accordingly.
Line 15: grammatical error.
Corrected.
Line 16: This is the first time you mention REMAP-CAP trial. Define this: "Randomized, Embedded, Multifactorial, Adaptive Platform trial for Community-Acquired Pneumonia (REMAP-CAP)".
This sentence was modified, as suggested.
Line16-18: Sentence is too long. Consider breaking it up into two sentences.
We agree and modified the text, accordingly.
Line 25-26: should be "During our study period, XX were male (55%) and XX were female (XX with an average age of 57 years (interquartile range, IQR, 44-69 years-old), and with median Charlson score of 4 (IQR, 1-4). Of the included patients, XX patients were diagnosed with diabetes, XX was HIV-positive. XX patients (88.9%) had central venous catheters in place." instead of "These patients were predominantly male (55%), median age was 57 years-old (interquartile range, IQR, 44-69 years-old), and median Charlson score was 4 (IQR, 1-4). Four patients had diabetes and one was infected with the HIV. Central venous catheters were in place for 88.9% (n=8) of patients."
We agree and modified the text, accordingly.
Line 28-30: This is a long sentence. Consider breaking it up in two sentences. If not, check the grammar. I suspect there is an inconsistency between plural vs. singular and verb conjugations.
This sentence was modified as other aspects in the text.
Table 1: Treatment and outcome column should be split in two different ones.
Other aspects were modified in table 1. Thank you.
Reviewer 2 Report
The manuscript by Riche et al. raises a fair point regarding the critical aspects potentially arising from the use of corticosteroids for the management of Covid-19 hospitalized patients, namely the increased risk to develop secondary fungal infections. The study presents an overview of the severe Covid-19 cases registered in two tertiary care hospitals in Porto Alegre, Southern Brazil between April 13th and August 31st 2020, in which high-dose therapy with corticosteroids was administered.
Major comments:
The topic is of extreme relevance, due to the current Covid-19 pandemic. Nonetheless, I believe that the manuscript should go trough significant modifications before being considered for publication. In fact, the title sets the expectation that a critical approach to the problem raised would lead – if not to an answer – at least to insights into how exactly the quest for an answer should be conducted. However, the authors did not draw any conclusions, besides a general “Physicians should be aware of the potential increase in the frequency of candidemia in patients with COVID-19 treated with corticosteroids”.
Moreover, this communication is significantly shorter and less organized than others published by this journal (see for comparison J. Fungi 2019, 5, 97; doi:10.3390/jof5040097). I believe that the manuscript would benefit from a more structured format, including at least a short introduction, the main body of the text, and conclusions. My main concern with this manuscript is the lack of an upfront systematic description of the patients’ cohort, which makes it difficult for the reader to understand what substantiates some of the claims that are made.
I also have some comments on the text specifically:
- The table could use more abbreviations, which could include the different underlying conditions, and the antifungal drugs. See for example J. Fungi 2020, 6, 211; doi:10.3390/jof6040211.
- “Since the beginning of the covid-19 pandemia we have observed a 10-fold increase in the frequency of candidemia in two tertiary care hospitals in Porto Alegre, Southern Brazil.” In my opinion a lot more information is required. Do the authors mean that the overall frequency of candidemia among all patients that were hospitalized increased 10 times? Or just the frequency of candidemia in the patients of a specific ward? Like infectious diseases? Or just in the ICU facility? Also, the increase is compared to the same time frame of the previous year? This information needs to be specific in order for the reader to be able to draw any conclusion.
- “All cases occurred between between April 13th and August 31st 2020, and followed the use of high-dose therapy with corticosteroids for severe covid-19 (Table 1)”. Does this mean that there were Covid-19 cases that were not treated with corticosteroids and did not develop candidemia? Or that some of them did, but were not included in the table because they were not treated with corticosteroids? The structure of the dataset is not clear to me. The authors should explain the features of the dataset more clearly.
- There are 9 patients displayed in the table, all of which developed Candidemia. Still, in the text the authors state: “Candidemia occurred in the intensive care for 6 out of the 9 patients (66.7%)”. Could the authors explain the discrepancy?
- “Overall mortality of fungal infection following steroid use in covid-19 was 77.7% (n=7)”. Again, grasping the significance of this is not easy. Does this mean that 8 out of 10 times that candidemia occurred in the presence of corticosteroid treatment the patient died? Are there cases in which covid patients did not require corticosteroid treatment but still develop candidemia? Also, what is the mortality rate of Covid-19 patients that did not develop candidemia but had symptoms serious enough to be admitted in the hospital?
- The authors link the candidemia with the corticosteroid use. However, a considerable proportion of COVID-19 critically ill patients develops acute respiratory distress and requires mechanical ventilation, which also predisposes them to nosocomial infections. Were these patients assisted in their respiration? What I am asking is: is the administration of corticosteroids the specific risk factor that facilitated candidemia in this case?
- “Higher corticosteroids doses might be associated with severe superinfections.”. I believe that there is nothing in the manuscript to substantiate this claim. What “higher doses” means? It needs to be more specific.
- The authors should spend more space in the text to comment and analyse the data presented in the table.
- It should also be clearly stated that this seems to be a very small cohort of patients to draw any general conclusion.
- On a final note, the authors should reference the relevant literature when claiming things like “The association between corticosteroids and fungal diseases is well known. Corticosteroids may promote fungal growth in vitro. Also, use of corticosteroids have been associated with increassed (TYPO, correct to INCREASED) risk for most serious fungal diseases including candidemia, invasive aspergillosis, fusariosis, and mucormycosis.”
Author Response
Reviewer 2
Comments and Suggestions for Authors
The manuscript by Riche et al. raises a fair point regarding the critical aspects potentially arising from the use of corticosteroids for the management of Covid-19 hospitalized patients, namely the increased risk to develop secondary fungal infections. The study presents an overview of the severe Covid-19 cases registered in two tertiary care hospitals in Porto Alegre, Southern Brazil between April 13th and August 31st 2020, in which high-dose therapy with corticosteroids was administered.
Major comments:
The topic is of extreme relevance, due to the current Covid-19 pandemic. Nonetheless, I believe that the manuscript should go trough significant modifications before being considered for publication. In fact, the title sets the expectation that a critical approach to the problem raised would lead – if not to an answer – at least to insights into how exactly the quest for an answer should be conducted. However, the authors did not draw any conclusions, besides a general “Physicians should be aware of the potential increase in the frequency of candidemia in patients with COVID-19 treated with corticosteroids”.
Moreover, this communication is significantly shorter and less organized than others published by this journal (see for comparison J. Fungi 2019, 5, 97; doi:10.3390/jof5040097). I believe that the manuscript would benefit from a more structured format, including at least a short introduction, the main body of the text, and conclusions. My main concern with this manuscript is the lack of an upfront systematic description of the patients’ cohort, which makes it difficult for the reader to understand what substantiates some of the claims that are made.
Thank you for your comment. We also feel the the title of the manuscript if a bit provocative, but that is the idea behid this report. We do believe this will attract readers for the text, and therefore the message should reach more easily the target audience. The manuscript was originally written to be presented as a letter, but we have modified the paper to be more like a concise communication. We appreciate that.
I also have some comments on the text specifically:
The table could use more abbreviations, which could include the different underlying conditions, and the antifungal drugs. See for example J. Fungi 2020, 6, 211; doi:10.3390/jof6040211.
We agree and modified the table according to the reference.
“Since the beginning of the covid-19 pandemia we have observed a 10-fold increase in the frequency of candidemia in two tertiary care hospitals in Porto Alegre, Southern Brazil.” In my opinion a lot more information is required. Do the authors mean that the overall frequency of candidemia among all patients that were hospitalized increased 10 times? Or just the frequency of candidemia in the patients of a specific ward? Like infectious diseases? Or just in the ICU facility? Also, the increase is compared to the same time frame of the previous year? This information needs to be specific in order for the reader to be able to draw any conclusion.
Thank you for pointing this out. We included incidence rates of candidemia in COVID-19 patients of each hospital for this period, and compared these with the incidence of candidemia in patients who did not have COVID-19, in the same period. Presenting this information we hope it became clearer for the reader.
“All cases occurred between between April 13th and August 31st 2020, and followed the use of high-dose therapy with corticosteroids for severe covid-19 (Table 1)”. Does this mean that there were Covid-19 cases that were not treated with corticosteroids and did not develop candidemia? Or that some of them did, but were not included in the table because they were not treated with corticosteroids? The structure of the dataset is not clear to me. The authors should explain the features of the dataset more clearly.
Yes, not all COVID-19 patients were treated with steroids. The use of corticosteroids became more frequent after the RECOVERY trial. And the majority of patients were not treated with high dose steroid therapy – as the patients presented in these trials. Considering the modifications in the paragraph, we hope that the information is now clear.
There are 9 patients displayed in the table, all of which developed Candidemia. Still, in the text the authors state: “Candidemia occurred in the intensive care for 6 out of the 9 patients (66.7%)”. Could the authors explain the discrepancy?
Considering the 9 patients, only 6 were admitted in the ICU at the time of the candidemia diagnosis.
We restructured the sentence for better understanding.
“Overall mortality of fungal infection following steroid use in covid-19 was 77.7% (n=7)”. Again, grasping the significance of this is not easy. Does this mean that 8 out of 10 times that candidemia occurred in the presence of corticosteroid treatment the patient died? Are there cases in which covid patients did not require corticosteroid treatment but still develop candidemia? Also, what is the mortality rate of Covid-19 patients that did not develop candidemia but had symptoms serious enough to be admitted in the hospital?
It was include the overall COVID-19 mortality of both institutions. It was also included a discussion in the manuscript.
The authors link the candidemia with the corticosteroid use. However, a considerable proportion of COVID-19 critically ill patients develops acute respiratory distress and requires mechanical ventilation, which also predisposes them to nosocomial infections. Were these patients assisted in their respiration? What I am asking is: is the administration of corticosteroids the specific risk factor that facilitated candidemia in this case?
Yes, we considered that the steroid therapy was and specific risk factor. This was a change in the profile of patients we used to have in these institutions.
“Higher corticosteroids doses might be associated with severe superinfections.”. I believe that there is nothing in the manuscript to substantiate this claim. What “higher doses” means? It needs to be more specific.
According to previous reference in literature, the use of corticosteroid was not recommended and high doses could be associated with adverse events. We added the reference for this statement.
The authors should spend more space in the text to comment and analyze the data presented in the table.
Please note that the manuscript is longer now in this revised version, in comparison to the original submission. This was intended to be a letter to the editor only and it is now transformed into a brief communication.
It should also be clearly stated that this seems to be a very small cohort of patients to draw any general conclusion.
We added a paragraph about limitations and the importance of the data presented in this study.
On a final note, the authors should reference the relevant literature when claiming things like “The association between corticosteroids and fungal diseases is well known. Corticosteroids may promote fungal growth in vitro. Also, use of corticosteroids have been associated with increassed (TYPO, correct to INCREASED) risk for most serious fungal diseases including candidemia, invasive aspergillosis, fusariosis, and mucormycosis.”
We added the reference for these statements.
Reviewer 3
Comments and Suggestions for Authors
This is a well-written letter. Tables/figures contribute substantially to content. Data have been correctly interpreted and conclusions are sound. However, a few points need of revision and elucidation before the article can be published.
Thank you for your comments.
Line 14: remove the sentence "Corticosteroids may promote fungal growth in vitro", which is not useful here.
We added the reference for this statement, as suggested by another reviewer. This is the basis for the link between steroids use and increased risk for fungal infections so we believe this is an useful statement.
Line 15: increased.
Corrected.
Line 20: the number of patients hospitalized during the study period is missing to do a prevalence of candidemia. We also need the usual number of candidiasis in the hospital to compare the periods.
We included incidence rates of candidemia in COVID-19 patients of each hospital for this period, and compared with the no COVID-19 patients in the same period. Also, it was superior in comparison to the previous year. Presenting this information we hope it became clearer for the reader.
Please note that the manuscript has been modified, in order to include that.
Line 28: Likewise the mortality of hospitalized patients during the period without fungal infection is missing to compare both death rate.
We included the overall mortality among COVID-19 patients in each institution for comparison.